# EEG-ImageNet: An Electroencephalogram Dataset and Benchmarks with Image Visual Stimuli of Multi-Granularity Labels

## Abstract

Exploring how brain activity translates into visual perception offers valuable insights into the biological visual system's representation of the world. Recent advancements have enabled effective image classification and high-quality reconstruction using brain signals obtained through Functional Magnetic Resonance Imaging (fMRI) or magnetoencephalography (MEG). However, the cost and bulkiness of these technologies hinder their practical application. In contrast, Electroencephalography (EEG) presents advantages such as ease of use, affordability, high temporal resolution, and non-invasive operation, yet it remains underutilized in related research due to a shortage of comprehensive datasets. To fill this gap, we introduce EEG-ImageNet, a novel EEG dataset featuring recordings from 16 participants exposed to 4000 images sourced from the ImageNet dataset. This dataset offers five times the number of EEG-image pairs compared to existing benchmarks. EEG-ImageNet includes image stimuli labeled with varying levels of granularity, comprising 40 images with coarse labels and 40 with fine labels. We establish benchmarks for both object classification and image reconstruction based on this dataset. Experiments with several commonly used models show that the best-performing models can achieve object classification with an accuracy around 60% and image reconstruction with two-way identification around 64%. These findings highlight the dataset's potential to enhance EEG-based visual brain-computer interfaces, deepen our understanding of visual perception in biological systems, and suggest promising applications for improving machine vision models.

## 1 Introduction

Recent advancements in reconstructing visual experiences from the human brain have seen significant progress, largely driven by the extensive use of functional magnetic resonance imaging (fMRI) (Heeger and Ress (2002); Logothetis (2008); Logothetis et al. (2001)) and magnetoencephalogram (MEG) (Benchetrit et al. (2023)) datasets. fMRI and MEG are widely used to investigate various cognitive functions, neurological disorders, and brain connectivity patterns (Antonello et al. (2024); Ye et al. (2024); Toneva et al. (2022); Tang et al. (2024)). Driven by the use of deep neural networks, particularly diffusion-based and transformer-based models, it is even possible to reconstruct human's visual perceptions from fMRI or MEG recordings (Takagi and Nishimoto (2023); Scotti et al. (2024); Ozcelik and VanRullen (2023); Cheng et al. (2023)). These models, with their powerful generalization and learning capabilities, have accelerated advancements in the field. This success in reconstructing visual objects from fMRI signals is largely attributed to the availability of large-scale datasets, which offer comprehensive data essential for conducting extensive studies and in-depth analyses (Richards et al. (2019); Lin et al. (2014)). For example, the Natural Scenes Dataset (Allen et al. (2022)), contains up to hundreds of thousands of high-quality natural image-fMRI pairs of 8 subjects, providing a solid data foundation for recent work in visual neuroscience. These models and large-scale datasets, in turn, have opened new avenues for understanding the brain's intricate functions and for developing advanced applications in brain-computer interfaces, neuroimaging, and beyond (St-Yves et al. (2023)).

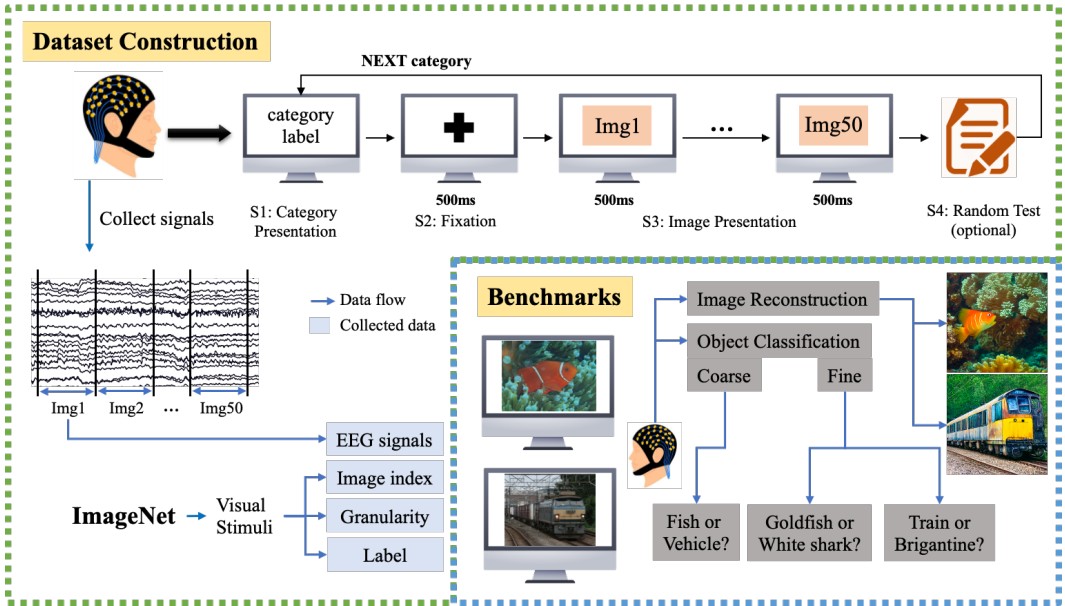

Figure 1: The overall procedure of our dataset construction and benchmark design. The experimental paradigm involves four stages: S1: Category Presentation (displaying the category label), S2: Fixation (500 ms), S3: Image Presentation (each image displayed for 500 ms), and S4: an optional random test to verify participant engagement. Each image presentation sequence includes 50 images from the given category, during which EEG signals are recorded. Data flow is indicated by blue arrows, while collected data is highlighted in gray. The stimuli images are sourced from ImageNet, with EEG signals aligned to image indices, granularity levels, and labels. The benchmarks (image reconstruction and object classification) are designed to evaluate coarse and fine granularity classification tasks.

In addition to fMRI and MEG, electroencephalography (EEG) is another vital tool in neuroscience research. EEG is easy to use, cost-efficient, and has high temporal resolution, making it a valuable tool for capturing rapid and real-time brain dynamics on the order of milliseconds (Teplan et al. (2002)). EEG signals can be obtained non-invasively by placing electrodes on the scalp, making it a less intrusive method for monitoring brain activity. These attributes position EEG as another promising modality for visual neuroscience research.

Although visual reconstruction has been achieved using fMRI and MEG, the high cost and inconvenience of these devices limit their widespread application in practical settings. In contrast, EEG presents advantages over both fMRI and MEG with its cost-efficient and portable features. However, studies on visual perception with EEG signals are limited because of two challenges: (1) the lack of large-scale, high-quality EEG datasets and (2) existing EEG datasets typically featured coarse-grained image categories, lacking fine-grained categories.

To the best of our knowledge, the most frequently used dataset is the data set provided by Spampinato et al. (2017), which involves 6 participants each watching 2000 image stimuli. However, this dataset's scale is smaller than existing fMRI datasets, limiting the possibilities for investigating neural aspects related to visual perception and developing deep learning models for relevant visual classification and reconstruction tasks. The limited data volume of current EEG datasets limits research findings' statistical power and generalizability.

On the other hand, the labels in existing EEG datasets are frequently coarse and lack the granularity needed for detailed analysis. Multi-granularity labels are essential because they allow for a more nuanced analysis at different levels of detail. For instance, labels can range from broad categories like "panda" or "golf ball" to more specific attributes like "Rottweiler" or "Samoyed". These challenges underscore the necessity for new, large-scale EEG datasets with high-quality, multi-granularity labels. Such datasets would enable researchers to explore the intricacies of visual processing with

greater accuracy and depth, facilitating advancements in both basic neuroscience and applied fields like brain-computer interfaces and clinical diagnostics.

To address these challenges, we present *EEG-ImageNet*, a novel EEG dataset specifically designed to promote research related to visual neuroscience, biomedical engineering, etc. As shown in Figure 1, EEG-ImageNet is a comprehensive dataset that includes EEG recordings from 16 subjects, each exposed to 4,000 images sourced from the ImageNet-21k (Ridnik et al. (2021)). These images span 80 different categories, with 50 images per category. The dataset is structured to support multi-granularity analysis, with 40 categories dedicated to coarse-grained tasks and 40 to fine-grained tasks. We further establish benchmarks for two primary tasks: object classification and image reconstruction. For the object classification task, we evaluated the dataset using several commonly used models, achieving a best accuracy of 60.88% on the 80-class classification. In the image reconstruction task, our experiments with advanced generative models, with the best model achieving a two-way identification of 64.67%. These benchmarks demonstrate the dataset's potential for advancing EEG-based research. By addressing the lack of large-scale, high-quality EEG datasets, EEG-ImageNet aims to drive EEG-based visual research forward, improve machine learning models, and provide deeper insights into visual perception and processing.

## 2 RELATED WORK

In this section, we review some datasets related to visual recognition and neuroscience and compare them with EEG-ImageNet, as shown in Table 1.

### 2.1 VISUAL RECOGNITION DATASET

Visual recognition is a cornerstone of computer vision, driven by datasets like ImageNet (Ridnik et al. (2021)), CIFAR (Krizhevsky et al. (2009)), and MS COCO (Lin et al. (2014)). ImageNet contains over 14 million annotated images, CIFAR-10/100 consist of thousands of 32x32 images in 10 and 100 classes, respectively, and MS COCO is known for its rich annotations supporting tasks such as object detection and segmentation.

Efforts to combine visual recognition with neuroscience have led to datasets like the SALICON dataset (Jiang et al. (2015)), which extends MS COCO with eye-tracking data, enabling the study of visual attention and saliency through large-scale annotations. Neuroscience datasets utilizing fMRI have further enriched this field. The BOLD5000 dataset (Chang et al. (2019)) includes fMRI data from subjects viewing 5000 images, aiding the exploration of visual perception. The Generic Object Decoding dataset (Horikawa and Kamitani (2017)) captures brain activity while subjects view and imagine objects, facilitating the decoding of mental images. The VIM-1 dataset (Kay et al. (2008)) from the study "Identifying Natural Images from Human Brain Activity" demonstrates the feasibility of decoding viewed images from brain activity. Additionally, the NSD (Allen et al. (2022)) is a large-scale fMRI dataset in visual neuroscience, recording high-resolution (1.8-mm) whole-brain 7T fMRI data from eight subjects exposed to 9,000–10,000 color natural scenes from the MS COCO dataset over the course of a year. Integrating neuroimaging and eye-tracking datasets with visual recognition tasks has opened new avenues for understanding how the brain interprets visual information, leading to insights and applications in brain-computer interfaces and neural decoding.

### 2.2 EEG DATASET

fMRI is renowned for its high spatial resolution, allowing researchers to obtain detailed images of brain activity by measuring changes in blood flow (Logothetis et al. (2001)). This capability makes fMRI particularly effective for identifying the specific brain regions involved in various cognitive and sensory tasks. In contrast, EEG offers several distinct advantages over fMRI. EEG is relatively easy to use and cost-efficient, with a straightforward setup that involves placing electrodes on the scalp to measure electrical activity. One of the most significant benefits of EEG is its exceptional temporal resolution, which captures neural dynamics on the order of milliseconds (Teplan et al. (2002)). This high temporal resolution makes EEG ideal for studying fast-occurring brain processes and real-time neural responses, providing insights into the timing and sequence of neural events (Liu et al. (2021)). EEG's non-invasive nature also makes it suitable for a wider range of participants.

Table 1: Detailed metadata for various neurological datasets based on visual stimuli.

| Dataset | #Subjects | Modalities | Visual Stimuli | #Stimuli | #Stimuli per Subject |
|---------|-----------|------------|----------------|----------|----------------------|
| SALICON (Jiang et al. (2015)) | - | Eye-tracking | MS COCO | 20,000 | - |
| BOLD5000 (Chang et al. (2019)) | 4 | fMRI | ImageNet, MS COCO, SceneUN (Xiao et al. (2010)) | 5,254 | 1,157-1,798 |
| GOD (Horikawa and Kamitani (2017)) | 5 | fMRI | ImageNet | 1,200 | 1,200 |
| VIM-1 (Kay et al. (2008)) | 2 | fMRI | Corel Stock Photo Library (Joshi and Guerzhoy (2017)) | 1,870 | 1,870 |
| NSD (Allen et al. (2022)) | 8 | fMRI, Eye-tracking | MS COCO | 9,000-10,000 | 22,000–30,000 |
| SEED (Zheng and Lu (2015)) | 15 | EEG | movie clips (4 min) | 15 | 15 |
| DEAP (Koelstra et al. (2011)) | 32 | EEG, ECG, … | music video (1 min) | 40 | 40 |
| AMIGOS (Miranda-Correa et al. (2018)) | 40 | EEG, ECG, … | long/short videos | 4+16 | 4+16 |
| Spampinato et al. (2017) | 6 | EEG | ImageNet | 2000 | 2000 |
| Things EEG1 (Grootswagers et al. (2022)) | 50 | EEG | Things (Hebart et al. (2019)) | 22248 | 22248 |
| Things EEG2 (Gifford et al. (2022)) | 10 | EEG | Things | 16740 | 16740 |
| EEG-SVRec (Zhang et al. (2024)) | 30 | EEG, ECG | short videos | 2636 | 121.9 |
| EIT-1M (Zheng et al. (2024)) | 5 | EEG | CIFAR-10 | 60,000 | 60,000 |
| Alljoined (Xu et al. (2024)) | 8 | EEG | MS COCO | 10,000 | 10,000 |
| **EEG-ImageNet** | 16 | EEG | ImageNet | 4,000 | 4,000 |

However, EEG signals collected using portable devices often have a low signal-to-noise ratio, which can complicate data analysis and reduce the accuracy of the results (Kannathal et al. (2005)).

Existing EEG datasets span a variety of research areas. The SEED (Zheng and Lu (2015)) focuses on emotion recognition with detailed EEG recordings from subjects exposed to various emotional stimuli. The BCI Competition IV datasets (Zhang et al. (2012)) provide EEG data for motor imagery tasks, while the TUH EEG Corpus (Shah et al. (2018)) is a large clinical EEG collection often used for benchmarking EEG data quality across different conditions.

In the realm of visual recognition, datasets like the EEG-Classification dataset (Spampinato et al. (2017)) involve 6 subjects viewing 2,000 images across 40 object classes from the ImageNet10k. The DEAP (Koelstra et al. (2011)) collects EEG and peripheral physiological signals from 32 participants as they watch 40 one-minute music videos, providing comprehensive emotional responses. Similarly, the AMIGOS (Miranda-Correa et al. (2018)) captures EEG and physiological responses from participants watching short video clips designed to evoke specific emotional states. The Things EEG (Grootswagers et al. (2022); Gifford et al. (2022)) utilizes in the RSVP paradigm features extremely short image presentation durations (50/100 ms) and collects EEG-image pairs from large-

scale datasets, making it a valuable resource for research in visual neuroscience. Another significant dataset is the EEG-SVRec (Zhang et al. (2024)), which includes EEG recordings from 30 participants interacting with short videos, aiming to capture detailed affective experiences.

In comparison, the EEG-ImageNet dataset offers several advantages. Firstly, its larger scale is conducive to training deep learning models. Secondly, the greater number of subjects facilitates inter-subject experiments and analyses. Additionally, the dataset features multi-granularity image labels, and the images are of high quality, selected from ImageNet21k and filtered to exclude small, blurry, or watermarked pictures.

## 3 DATASET CONSTRUCTION

During the data collection process of our user study, participants are presented with a visual stimuli dataset containing 4000 natural images from ImageNet. Throughout this process, we continuously record their EEG signals. The whole experimental process is carried out in the laboratory environment. This section describes the entire process of EEG-ImageNet dataset construction. The detailed code can be accessed openly through the url `https://anonymous.4open.science/r/EEG-ImageNet-Dataset-anonymous-237A`.

### 3.1 ETHICS AND PRIVACY

To protect participants' privacy and physical health, our user study adheres to strict ethical guidelines for human research, with approval from the ethics committee[1]. The study has undergone a comprehensive ethical review to safeguard participants' rights. In accordance with ethical standards, we have taken several steps to protect participants' privacy, including data anonymization and obtaining informed consent from all participants. Additionally, participants are thoroughly informed about the study's objectives, procedures, and potential outcomes. The EEG data collection method employed in this research is non-invasive and poses no risk to participants. This approach ensures compliance with ethical standards while maintaining the integrity of the research findings.

### 3.2 PARTICIPANTS

We enlist a total of 16 participants via social media, including 10 males and 6 females. These participants are all college students aged between 21 and 27, with an average age of 24.06 and a standard deviation of 1.69. Their majors encompass computer science, mechanical engineering, chemistry, and environmental engineering, and they range from undergraduate to postgraduate levels. All participants are right-handed and assert their proficiency in utilizing image search engines in their daily routines. Each participant dedicates approximately 2 hours to complete the experiment, which includes 30 minutes for equipment setup and task instructions. Before the experiment, participants are informed of a compensation of US$11.8 per hour upon completion, to ensure the quality of the data collected for the study.

### 3.3 STIMULI DATASET

The dataset used for visual stimuli was a subset of ImageNet21k, containing 80 categories of objects. Each category comprises 50 manually curated images, ensuring that each image has a width and height greater than 300 pixels and prominently features an object corresponding to its class label in ImageNet. Additionally, every image is free of watermarks. In this manner, we have selected a total of 4000 high-quality natural images as our visual stimulus dataset.

Among all categories, the first half is consistent with the EEG-Classification dataset (Spampinato et al. (2017)), comprising 40 significantly distinct categories from ImageNet1k. We treat these as *coarse-grained* tasks. The latter 40 categories are designed as a *fine-grained* task, divided into 5 groups with 8 categories each. The categories within the same group share the same parent node in WordNet, and each category label is either a leaf node or a sub-leaf node in WordNet. This selection ensures that the chosen categories represent similar granularity while avoiding overly obscure categories, thereby minimizing potential biases in the experimental results. For instance, coarse-grained

---

[1] detailed information will be released after review

categories include items such as African elephants, pandas, mobile phones, golf balls, bananas, and pizzas. Under the parent node "musical instruments," the fine-grained categories include accordions, cellos, flutes, oboes, snare drums, and trombones. Detailed information about all the visual stimuli categories and their respective WordNet IDs can be found in our GitHub repository.

## 3.4 PROCEDURE

A pilot study can ensure the correctness of the overall experimental process and the reliability of the acquisition equipment. We conduct a pilot study on two participants whose data are not included in the final analysis to determine hyperparameters of the experimental design, such as font size, fixation time, number of categories, etc.

Before engaging in the user study, participants are required to fill out an entry questionnaire and sign a consent about the protection of privacy security. They will receive an orientation regarding the primary tasks and operational procedures. Additionally, they will be notified of their right to withdraw from the study at any point. Before the main trials, participants will undergo a series of training trials designed to acquaint participants with the procedures of the formal experiments.

Every participant is required to select a random seed before the experiment to randomize the order of the categories. This randomization guarantees a fair distribution of categories and images among participants. The experimental platform follows a sequential and repetitive process as illustrated in Figure 1. (S1) The experimental platform presents the current category label. Participants can proceed to the next stage by pressing the space key. (S2) A fixation cross is shown at the center of the screen, ensuring attention is drawn when images are displayed. This fixation period lasts for 500 ms. (S3) The 50 images of this category are sequentially presented using the Rapid Serial Visual Presentation (RSVP) paradigm, which is commonly employed in psychological experiments. Each image is presented for a duration of 500 ms (Kaneshiro et al. (2015)). (S4) Random tests are conducted to verify the participant's engagement in the experiment after the presentation. Data from categories for which participants fail the test will not be included in final analyses. The EEG signals of the participant will be captured and recorded continuously during the entire process. The program will cycle back to step S1 and display the next category, repeating this process until all the images have been presented.

Ultimately, the dataset we construct includes the EEG signals of participants exposed to each image visual stimulus in each valid session, along with the corresponding category's wnid and the image's index in ImageNet21k.

## 3.5 DATASET DESCRIPTION

The EEG-ImageNet dataset contains a total of 63,850 EEG-image pairs from 16 participants. Each EEG data sample has a size of (n_channels, $f_s \cdot T$), where n_channels is the number of EEG electrodes, which is 62 in our dataset; $f_s$ is the sampling frequency of the device, which is 1000 Hz in our dataset; and T is the time window size, which in our dataset is the duration of the image stimulus presentation, i.e., 0.5 seconds. Due to ImageNet's copyright restrictions, our dataset only provides the file index of each image in ImageNet and the wnid of its category corresponding to each EEG segment. Additional information about the dataset is shown in Appendix A.1.

## 4 BENCHMARKS

In this section, we detail the benchmarks of our study by outlining the preprocessing steps, feature extraction methods, task definitions, and models used.

## 4.1 PREPROCESSING

We perform a series of preprocessing steps for the raw EEG data we collect to eliminate noise and artifacts and improve signal quality (Ye et al. (2024)). The preprocessing pipeline includes the following stages: First, re-referencing: Re-referencing is done using the offline linked mastoids method, which uses the average of the M1 and M2 mastoid electrodes as the new reference point (Yao et al. (2019)). This minimizes potential biases and improves the signal-to-noise ratio. Then, filtering:

Filtering is performed using a 0.5 Hz to 80 Hz band-pass filter to remove low-frequency drifts and high-frequency noise. Additionally, 50 Hz environmental noise is eliminated. Finally, artifact removal: Artifact removal eliminates abnormal amplitude signals and artifacts caused by blinks or head movements.

## 4.2 FEATURE EXTRACTION

In our benchmarks, for models requiring time-domain signals as direct input, we extract the 40ms-440ms segment of each EEG signal as the feature input. This approach helps to minimize the influence of preceding and subsequent image stimuli on the current stimulus. For models requiring frequency-domain features as input, we extract the differential entropy (DE) of the extracted time-domain signals as features, as this characteristic effectively captures the complexity and variability of brain activity in the frequency domain (Duan et al. (2013)). According to the general division in neuroscience, the frequency bands are categorized as delta (0.5-4 Hz), theta (4-8 Hz), alpha (8-13 Hz), beta (13-30 Hz), and gamma (30-80 Hz). We use the Welch method with a sliding window to estimate the power spectral density $P(f)$ in each frequency band. Then, we normalize the data and calculate the differential entropy (DE) using the formula, $DE = -\int P(f)\log(P(f))df$. Consequently, for each segment of EEG signals, we obtain the differential entropy (DE) for each electrode and each frequency band.

## 4.3 TASK DEFINITION

In our benchmarks, we test our dataset on two tasks: *object classification* and *image reconstruction*. The object classification task aims to classify the category of the corresponding image stimulus the participant is exposed to with their EEG signals. We evaluate the models using classification accuracy. In the image reconstruction task, given a specific EEG segment, the goal is to reconstruct the image stimulus the participant is exposed to. We evaluate the generated results using two-way identification (Scotti et al. (2024)) under different visual neural networks (refer to Appendix A.3 for details.).

On our multi-granularity labeled image dataset, we test various tasks with different levels of granularity. Additionally, due to the inherent significant inter-individual differences in EEG signals, all models in our benchmarks are trained exclusively in an intra-subject experimental setup. Furthermore, to mitigate the significant temporal effects (Li et al. (2020)) observed in the EEG-Classification dataset, all our experimental setups strictly adhere to a dataset split where the first 30 images of each category are used as the training set, and the last 20 images are used as the test set. This approach ensures that the model learns features that are less correlated with irrelevant information. We strongly recommend that all researchers using the EEG-ImageNet dataset adopt a similar dataset split methodology. We also explain the impact of this split in Appendix A.5.

## 4.4 MODELS

In the object classification task, we employ simple machine-learning classification models such as ridge regression, KNN, random forest, and SVM. Additionally, we implement deep learning models including MLP, EEGNet (Lawhern et al. (2018)), and RGNN (Zhong et al. (2020)). These models are the most commonly used and have demonstrated excellent performance in EEG-related research, which is why we have selected them for our study. MLP consists of two hidden layers, while EEGNet and RGNN are implemented using their original architectures. We use cross-entropy as the loss function. We train the models for each participant for 2000-3000 epochs on an NVIDIA 4090 GPU.

In the image reconstruction task, we use a Stable Diffusion 1.4 (Rombach et al. (2022)) as the backbone and train a two-layer MLP as an encoder to generate the prompt embeddings for input. For our reconstructions, we use 50 denoising timesteps with PNDM noise scheduling to generate 512x512 images. We use MSE as the loss function to align the encoder's output with the CLIP (ViT-L/14) (Radford et al. (2021)) embeddings of the captions of real images obtained through BLIP (Li et al. (2022)). The model for each participant is trained for 2000 epochs with on an NVIDIA 4090 GPU.

For model and training details, please refer to the open-source code and Appendix A.3.

Table 2: The average results of all participants in the object classification task. * indicates the use of time-domain features, otherwise it indicates the use of frequency-domain features. † indicates that the difference compared to the best-performing model is significant with p-value ¡ 0.05.

| Model | | Acc (all) | Acc (coarse) | Acc (fine) |
|---|---|---|---|---|
| Classic model | Ridge | 0.286±0.074† | 0.394±0.081† | 0.583±0.074† |
| | KNN | 0.304±0.086† | 0.401±0.097† | 0.696±0.068† |
| | RandomForest | 0.349±0.087† | 0.454±0.105† | 0.729±0.072† |
| | SVM | **0.392±0.086†** | **0.506±0.099†** | **0.778±0.054†** |
| Deep model | MLP | 0.404±0.103† | **0.534±0.115** | **0.816±0.054** |
| | EEGNet* | 0.260±0.098† | 0.303±0.108† | 0.365±0.095† |
| | RGNN | 0.405±0.095† | 0.470±0.092† | 0.706±0.073† |
| | LSTM | 0.356±0.082† | 0.459±0.101† | 0.745±0.068† |
| | Transformer | 0.367±0.085† | 0.460±0.108† | 0.750±0.070† |
| | EEGConformer | **0.416±0.109** | 0.519±0.108 | 0.801±0.067 |
| **Model** | | **F1 (all)** | **F1 (coarse)** | **F1 (fine)** |
| Classic model | Ridge | 0.261±0.070† | 0.373±0.082† | 0.610±0.121† |
| | KNN | 0.286±0.081† | 0.380±0.096† | 0.717±0.132† |
| | RandomForest | 0.323±0.083† | 0.425±0.099† | 0.723±0.092† |
| | SVM | **0.378±0.083†** | **0.486±0.105†** | **0.770±0.054†** |
| Deep model | MLP | 0.397±0.100† | **0.523±0.108** | **0.819±0.053** |
| | EEGNet* | 0.251±0.095† | 0.291±0.098† | 0.374±0.102† |
| | RGNN | 0.401±0.098† | 0.455±0.087† | 0.723±0.079† |
| | LSTM | 0.347±0.076† | 0.437±0.092† | 0.729±0.058† |
| | Transformer | 0.353±0.079† | 0.451±0.096† | 0.761±0.075† |
| | EEGConformer | **0.413±0.102** | 0.513±0.101 | 0.812±0.070 |

## 5 EXPERIMENTS, RESULTS

In this chapter, we present the experimental results of our study, focusing on two benchmark tasks: object classification and image reconstruction.

### 5.1 OBJECT CLASSIFICATION

In the object classification task, we conduct experiments at three levels of granularity: all, coarse, and fine. The "all" category represents the 80-class classification accuracy, the "coarse" category represents the 40-class classification accuracy, and the "fine" category represents the average accuracy of five 8-class classification tasks. The performance of each model is detailed in Table 2, which shows the average results of all participants.

The table shows that RGNN achieves the highest accuracy in the "all" task among all models, with an 80-class classification accuracy reaching 40.50%. MLP slightly lags behind RGNN in the "all" task accuracy, but it significantly outperforms in the "coarse" and "fine" tasks, with a 40-class accuracy of 53.39% and an average 8-class accuracy of 81.63%. Among the classical machine learning models, SVM exhibits the highest classification performance, only slightly trailing RGNN and MLP across the three different granularity scenarios. The lower accuracy of RGNN in the "fine" task might be due to the complex model structure, which may require more fine-tuning to better adapt to easier tasks. Meanwhile, the lower performance of time-domain features compared to frequency-domain features suggests that frequency-domain features might be more informative for the classification tasks in this study. Specific performance details for each participant in each task are provided in Appendix A.4. We find that the ranking of participants' accuracy is relatively consistent across different models.

We compile the accuracy of each participant for the "all" task under the SVM, MLP, and RGNN models, as shown in Figure 2a. We observe significant differences between participants, indicating high variability in individual responses. The best-performing participant achieves an accuracy of

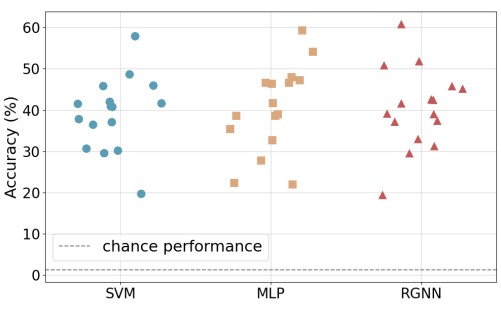
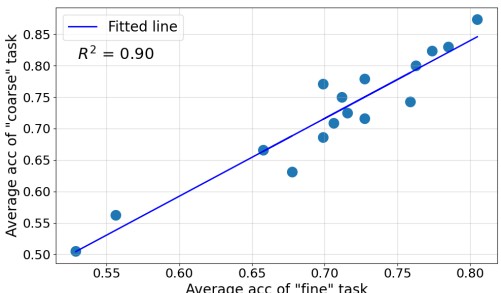

(a) Classfication accuracy for each participant in the object classification task (all) across models.

(b) Classification accuracy of the "coarse" (vertical axis) and "fine" task (horizontal axis).

Figure 2: Classification performance of the object classification, each dot represents the classification performance of a single participant.

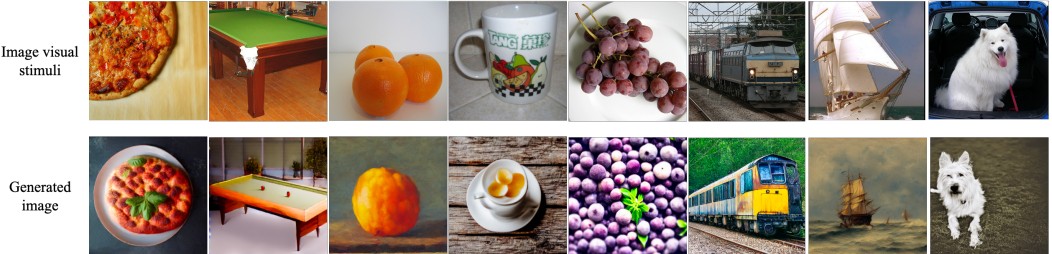

Figure 3: The image reconstruction results of a single participant (S8).

60.88%. To better compare the differences between "fine" and "coarse" tasks in EEG-ImageNet, we modify the coarse-grained task. We randomly select 8 coarse-grained categories and use the RGNN model for training and testing. This process is repeated 5 times, and the average accuracy for each participant is calculated and plotted alongside their average "fine" task accuracy in Figure 2b. We then perform linear regression on the data points, and the resulting function has a slope greater than 1, indicating that participants generally achieve better results on the "coarse" classification tasks. This finding is also consistent with intuition.

## 5.2 IMAGE RECONSTRUCTION

Figure 3 shows some of the results from our image reconstruction pipeline of a single participant (S8). From the selected images, it can be seen that the reconstruction pipeline can effectively restore the category information of the image stimuli. However, restoring low-level details such as color, position, and shape is inaccurate. This may be because we align the EEG signals to the image captions obtained from BLIP. Due to the limited descriptive precision of BLIP, we cannot fully leverage the diffusion model's generative capabilities.

Table 3 shows the average two-way identification (chance=50%) of the images generated by our reconstruction pipeline using different visual neural networks. Two-way identification refers to comparing the generated image with the original and one distractor to evaluate accuracy. The highest accuracy achieved by CLIP suggests that integrating vision transformers with extensive pre-training on diverse image-text data can significantly enhance model performance for complex tasks like image reconstruction from EEG signals. Additional results can be found in Appendix A.4.

Table 3: The average results of all participants in the image reconstruction task.

| Method | Alex(2) | Alex(5) | Incep | CLIP(ViT-L/14) | Eff | SwAV |
|---|---|---|---|---|---|---|
| Metrics | 56.05% | 62.99% | 56.75% | **64.67%** | 0.867 | 0.597 |

## 6 DISCUSSION AND CONCLUSION

In this paper, we introduced EEG-ImageNet, a novel EEG dataset designed to advance research in visual neuroscience. EEG-ImageNet is comprehensive and richly annotated, comprising EEG recordings from 16 subjects exposed to 4000 images from 80 different categories. We established benchmarks for two primary tasks using EEG-ImageNet: object classification and image reconstruction. The introduction of EEG-ImageNet and the established benchmarks provide a robust foundation for subsequent research. We outline the limitations of our dataset and explore how it could guide future research efforts to advance machine learning and brain-computer interface design.

**Limitation**. Firstly, while our dataset is more comprehensive than similar works, each participant's data is still relatively limited. This necessitates the development of inter-subject models to overcome this limitation and enhance generalizability. Additionally, it is limited in representation, as participants were drawn from a convenience sample at our university. This results in an age distribution skewed towards college-aged individuals and a racial composition predominantly White and Asian, which limits the dataset's generalizability. Future work should aim to include a more diverse and extensive participant pool. Secondly, while we employed methods such as reducing the segment length of each EEG recording and sequentially splitting the training and test sets to mitigate the temporal effect, we were unable to eliminate it completely. Future work should explore more sophisticated techniques to address this issue. Lastly, our benchmarks did not incorporate many of the latest deep-learning methods. We believe that recent advancements in deep learning could greatly benefit from our comprehensive dataset, potentially leading to significant breakthroughs in visual neuroscience.

**Insight for ML**. The EEG-ImageNet dataset provides a comprehensive resource for developing models in visual recognition tasks, enabling the development of sophisticated deep-learning models capable of capturing intricate patterns within EEG data. Future research could leverage the dataset to enhance domain adaptation and transfer learning techniques, facilitating effective inter-subject task completion. Researchers might develop state-of-the-art models for the benchmarks we defined, or even create new tasks. By offering a diverse set of visual stimuli and supporting multi-level classification tasks, EEG-ImageNet could foster the creation of hierarchical models that mirror human cognitive processes and improve the generalization capabilities of machine learning algorithms.

**Insight for BCI**. As hardware technology progresses, portable EEG devices are becoming increasingly feasible, offering new opportunities for real-time BCI applications. Researchers could use the dataset to develop robust BCI systems that accurately interpret user intent from EEG signals. The comprehensive size and diverse visual stimuli in EEG-ImageNet allow for the creation of adaptive BCI systems that learn and respond to individual user patterns. This paves the way for personalized neurotechnology solutions, particularly enhancing human-computer interaction for individuals with disabilities. Furthermore, addressing privacy protection and ethical concerns will be crucial as BCI technology advances, ensuring user data is securely handled and individual rights are respected.

**Ethics Statement**. Our dataset was collected from human participants, making privacy protection and ethical use of the data a priority. The user study received approval from the institutional ethics committee, and all participants provided informed consent, agreeing to the public release of the anonymized data for research purposes. It is important to note that our dataset exhibits certain biases, including racial bias, which was influenced by the demographics of available participants. Researchers using this dataset should be aware of this limitation and consider it in their analysis. Additionally, all participants were right-handed. This decision was made to control for variability, as previous research has shown significant differences in EEG signals between right- and left-handed individuals Provins and Cunliffe (1972). By selecting only right-handed participants, we aimed to ensure the consistency and reliability of the EEG signals across subjects. Researchers should consider this factor when conducting studies where handedness is not a variable of interest to minimize variability in the results.

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

## A  APPENDIX

### A.1  ADDITIONAL INFORMATION ABOUT DATASET

The specific statistics of the dataset are shown in Table 4.

As shown in Listing 1, the EEG-ImageNet dataset storage format is provided after review. The dataset can be accessed through the cloud storage link available in our GitHub repository. Due to file size limitations on the cloud storage platform, we split the dataset into two parts: "EEG-ImageNet_1.pth" and "EEG-ImageNet_2.pth", each containing data from 8 participants. Users can choose to use only one of the parts based on their specific needs or device limitations. Demographic information is also provided at the file level.

Table 4: The Statistics of EEG-ImageNet Dataset.

|  | #Categories | #Images | #Subjects | #EEG-image pairs | Datasize |
|---|---|---|---|---|---|
| EEG-ImageNet | 80 | 4000 | 16 | 63850 | 15.88GB |

```
{
    "dataset": [
        {
            "eeg_data": torch.tensor,
            "granularity": "coarse"/"fine",
            "subject": 15,
            "label': 'ne2106550',
            "image': 'n02106550_1410.JPEG'
        }, ...
    ],
    "labels": [
        "n02106662", ...
    ],
    "images": [
        "n02106662_13.JPEG", ...
    ]
}
```

Listing 1: EEG-ImageNet dataset format.

## A.2 APPARATUS

All the image stimuli are presented on a desktop computer that has a 27-inch monitor with a resolution of 2,560×1,440 pixels and a refresh rate of 60 Hz. Participants are required to use the keyboard to interact with the platform. EEG signals are captured and amplified using a Scan NuAmps Express system (Compumedics Ltd., VIC, Australia) and a 64-channel Quik-Cap (Compumedical NeuroScan). A laptop computer functions as a server to record EEG signals and triggers using Curry8 software. Throughout the experiment, electrode-scalp impedance is maintained under $50\Omega$, and the sampling rate is set at 1,000Hz.

## A.3 EXPERIMENTAL SETUP DETAILS

In the object classification task, we conduct experiments under three different granularity settings: the "all" task includes all 80 categories; the "coarse" task includes 40 coarse-grained categories; and the "fine" task includes 8 fine-grained categories that belong to the same parent node, with the average accuracy calculated across 5 groups. Each classification model maintain parameter consistency across the three tasks. We train one model per participant, ensuring that the parameters are consistent between models for different participants as well.

The model structures and hyperparameters are as follows. For SVM, we try linear, polynomial, and radial basis function (RBF) kernels. The regularization parameter is tested from values $\{10^{-3}, 10^{-2}, 10^{-1}, 1, 10^1, 10^2, 10^3\}$. For RandomForest, we try to set the number of trees in the forest from values $\{20, 50, 100, 200, 500\}$, with all other parameters set to their default values. For KNN, we set the number of neighbors to $\{5, 10, 15, 20\}$. For ridge regression, all parameters are set to their default values. For RGNN, when calculating the edge weights between electrodes, we use the hardware parameters of our data collection device to determine the topological coordinates of each electrode. In addition to the standard implementation, we add two batch normalization layers. The main hyperparameters adjusted are the number of output channels of the graph convolutional network (i.e., the hidden layer dimension) and the number of hops (i.e., the number of layers). These are set to $\{100, 200, 400\}$ and $\{1, 2, 4\}$ respectively. For EEGNet, we use the standard implementation and set the length of the first step convolution kernel to half the number of sampling time points, which is 200. The main hyperparameters adjusted are the number of output channels for the first convolutional layer (F1) and the depth multiplier (D), which are set to $\{8, 16, 32\}$ and $\{2, 4, 8\}$

respectively. For MLP, we set two hidden layers with dimensions of 256 and 128, respectively. Each linear layer is followed by a batch normalization layer and a dropout layer with a probability of 0.5.

For all deep models, we use the cross-entropy loss function. In MLP and EEGNet, we use the SGD optimizer with learning rate $10^{-3}$, weight decay $10^{-3}$, and momentum 0.9, training for 2000 epochs. After that, we adjust the learning rate to $10^{-4}$ and weight decay to $10^{-4}$ and continue training for another 1000 epochs. In RGNN, we use the Adam optimizer with learning rate $10^{-3}$ and weight decay $10^{-3}$, training for 2000 epochs. Subsequently, we adjust the learning rate to $10^{-4}$ and weight decay to $10^{-4}$ and train for an additional 1000 epochs. The batch size is uniformly set to 80.

In the image reconstruction task, we first use blip-image-captioning-base to obtain captions for each stimulus image, with parameters set to "max length": 200 and "num beams": 20 to achieve relatively detailed descriptions. Then, we use the CLIP ViT-L/14 version's tokenizer and text encoder to compute the CLIP embedding for each caption, resulting in dimensions of (77, 768). Next, we map the frequency domain features of the EEG signals to (77, 768) using an MLP with two hidden layers, having dimensions of 1024 and 2048, respectively. This output is then fed into a frozen Stable Diffusion 1.4 model built with the diffusers library. After 50 steps of inference, we obtain image outputs with dimensions of 512x512. The Stable Diffusion model we built specifically consists of the UNet2DConditionModel and AutoencoderKL, and we use PNDMScheduler as the noise scheduler.

Two-way identification refers to a method where the generated image is compared with the original image and one distractor image. The task is to correctly identify the original image from the pair. This method evaluates the ability of the reconstruction pipeline to produce images that are distinguishable and recognizable as the original stimuli. In this study, we selected the second and fifth convolutional layers of AlexNet, the output before the linear layer of Inception, and the embeddings from CLIP ViT-L/14 as comparison features to calculate two-way identification.

All the implementations mentioned above are open-sourced and available in the GitHub repository.

## A.4 ADDITIONAL EXPERIMENTAL RESULTS

Table 5 shows the performance of the best-performing participant across all models and tasks.

Table 5: The best results of all participants in the object classification task.

| | Model | Acc (all) | Acc (coarse) | Acc (fine) |
|---|---|---|---|---|
| | Ridge | 0.4550 | 0.5375 | 0.7200 |
| | KNN | 0.5025 | 0.6063 | 0.8013 |
| Classic model | RandomForest | 0.5006 | 0.6488 | 0.8450 |
| | SVM | **0.5794** | **0.7038** | **0.8588** |
| | RGNN | **0.6088** | 0.6525 | 0.8050 |
| Deep model | EEGNet* | 0.4413 | 0.5213 | 0.5988 |
| | MLP | 0.5925 | **0.7413** | **0.8875** |

Figure 4 shows the accuracy for each participant in the object classification task across SVM, MLP, and RGNN models. We find that the ranking of participants' accuracy is relatively consistent across different models.

Figure 5 presents more image generation results selected from other participants, with Figure 5a showing good cases and Figure 5b showing bad cases. We identified **three** main types of bad cases. Similar to the first two images, the reconstructed images lack or misrepresent low-level information such as color and shape. These errors are relatively common and are due to the limitations of our feature mapper and the simple structure of the reconstruction pipeline, resulting in insufficient information restoration. Similar to the latter two images, the reconstructed images lack detail. This limitation is due to the number of denoising steps in the diffusion model and the inherently low signal-to-noise ratio of EEG signals.

We also observed that for certain categories, especially fine-grained ones, all test data points resulted in near-noise outputs, which drew our attention. When we directly input category labels as text prompts into Stable Diffusion 1.4, we found that the generated images had poor realism and three-

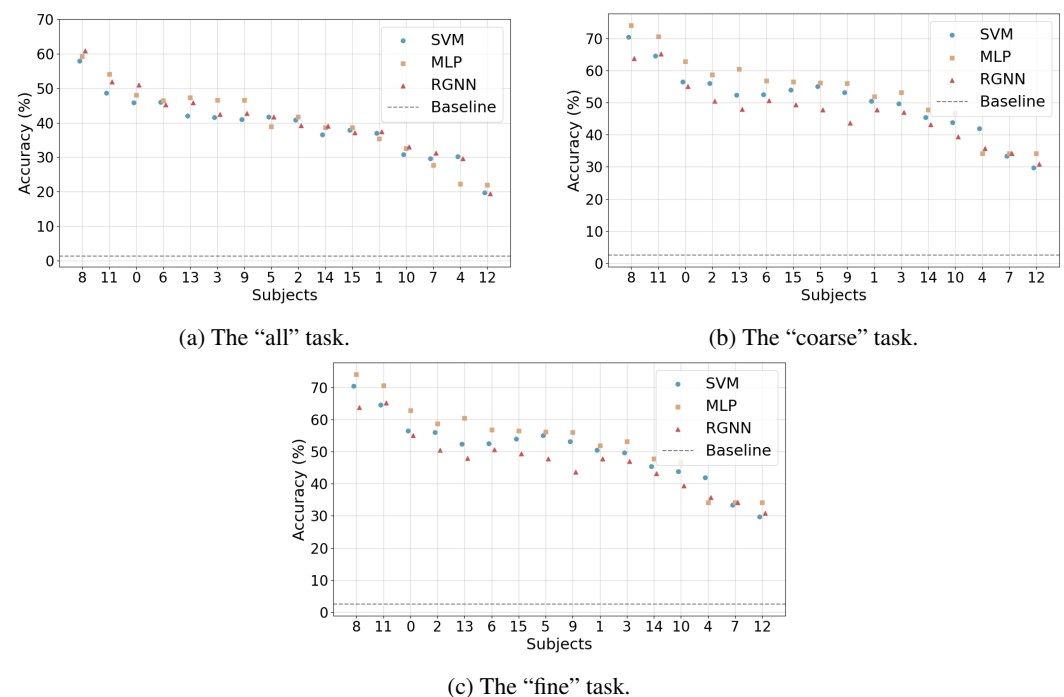

(a) The "all" task.

(b) The "coarse" task.

(c) The "fine" task.

Figure 4: Acc for each participant in the object classification task across SVM, MLP, and RGNN Models.

dimensional structure. Figure 5c compares these images with those generated by our reconstruction pipeline from the training set. This improvement suggests that we can use EEG, which can be quickly and extensively obtained as human feedback signals, to enhance the performance of text-image pre-trained models or generative models. This will be the direction of future research.

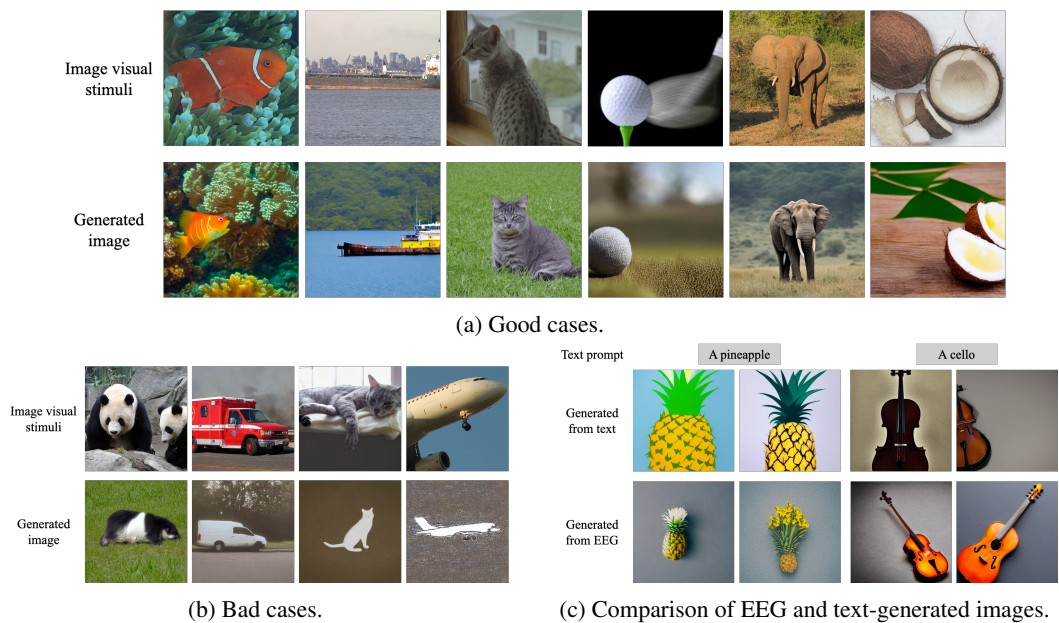

(a) Good cases.

(b) Bad cases.

(c) Comparison of EEG and text-generated images.

Figure 5: More results in the image generation task.

## A.5 TEMPORAL EFFECT

In our experimental paradigm, to reduce the cognitive load on participants, we group images of the same category together and use the RSVP (Rapid Serial Visual Presentation) paradigm for continuous rapid stimulation. This approach may cause the model to learn temporal continuity features rather than the intrinsic characteristics of the category stimuli. In this study, we use narrower cropped time segments as input features and divide the training and test sets based on temporal order to reduce the impact of temporal effects. In Figure 6, we plotted the average classification accuracy for images at different index positions in the test set under various training and test set splits.

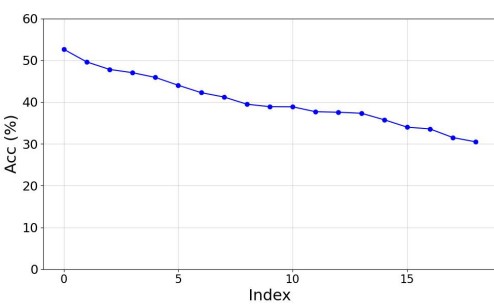 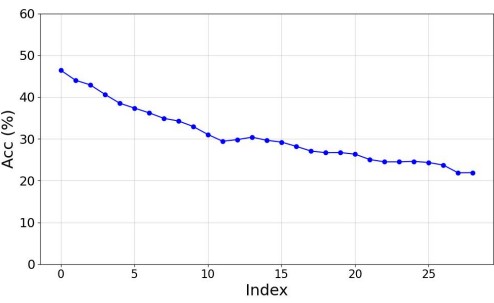

(a) 30/20 split for the training and test sets.   (b) 20/30 split for the training and test sets.

Figure 6: Average classification accuracy under different training and test set splits, with accuracy plotted against the indices of image stimuli in the test set.

We observed that the first few images in the test set have significantly higher accuracy, indicating a strong temporal effect, while the accuracy tends to stabilize for the subsequent images. Next, we also used several statistical methods for analysis. For the 30-20 split, we first calculated the sliding window standard deviation to measure local volatility. With a sliding window size of 5, the standard deviation at the 14th data point was less than 0.01, indicating a **convergence** trend at this point. We then checked the stationarity of the data using the ADF test, obtaining an ADF statistic of -1.1874 and a p-value of 0.6790, suggesting that the data might not have fully converged. For the 20-30 split, the standard deviation at the 16th data point was less than 0.01 with a sliding window size of 5. The ADF test yielded a statistic of -3.8505 and a p-value of 0.0024, indicating that the data is **stationary and convergent**.

Also, In our benchmark experiments, we employed a **narrower temporal segmentation** by extracting the 40ms-440ms to minimize the impact of temporal effects within the RSVP paradigm. Further, our dataset publicizes the **time series information** for each EEG segment along with the corresponding image presentation order. This information enables classification experiments to better address temporal effects by performing the data partitioning. As such, our dataset serves as a valuable benchmark, and we hope that our dataset can contribute significantly to future research aimed at mitigating temporal effects in the RSVP paradigm.

In RSVP paradigms, temporal effects are inherently present due to the sequential presentation of stimuli. Employing a fully randomized sampling method can exacerbate this issue, as the preceding and succeeding images introduce temporal effects that impact the semantic alignment of brain signals for a given image. To address this, we adopted a refined experimental design, as illustrated in Figure 7, to ensure consistent EEG responses corresponding to the same image semantics. Our two-phase experiments separated training and testing stages, avoiding block design overlaps. This separation ensures no temporal dependencies between experimental conditions, enabling robust signal decoding. Thus far, data from three participants have been collected, with four more participants scheduled for data collection. Importantly, the data acquisition methods and formats remain consistent with those of the original dataset, ensuring compatibility and facilitating integration for analysis. The experiment will result in a total of 14 additional hours of recordings and 28,000 EEG-image pairs, with preliminary results from the new dataset summarized in Table 6.

Comparing the results of participants in Stage 2 with those in the original experiment, we observe a decline in classification performance across different models and granularity levels. This decrease can be attributed to the temporal effect inherent in the block design experimental paradigm. How-

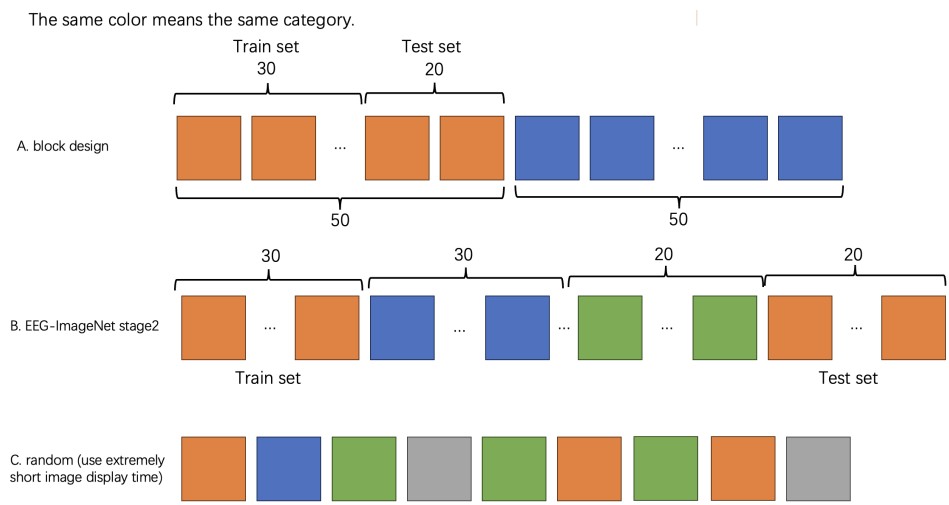

Figure 7: The comparison of block design, random design and Stage2 design of EEG-ImageNet.

Table 6: The average results of all participants in the object classification task of Stage2. * indicates the use of time-domain features, otherwise it indicates the use of frequency-domain features. † indicates that the difference compared to the best-performing model is significant with p-value ¡ 0.05.

| Model | | Acc (all) | Acc (coarse) | Acc (fine) |
|---|---|---|---|---|
| Classic model | Ridge | 0.143±0.043† | 0.211±0.060† | 0.395±0.099† |
| | KNN | 0.176±0.078† | 0.265±0.101† | 0.533±0.116† |
| | RandomForest | 0.223±0.091† | 0.312±0.111† | 0.560±0.128† |
| | SVM | **0.235±0.076** | 0.328±0.097† | 0.614±0.132† |
| Deep model | MLP | 0.249±0.080 | 0.356±0.105 | **0.676±0.145** |
| | EEGNet* | 0.129±0.038† | 0.199±0.082† | 0.357±0.088† |
| | RGNN | **0.256±0.088** | **0.360±0.098** | 0.651±0.132 |
| **Model** | | **F1 (all)** | **F1 (coarse)** | **F1 (fine)** |
| Classic model | Ridge | 0.134±0.042† | 0.205±0.060† | 0.393±0.098† |
| | KNN | 0.169±0.075† | 0.257±0.096† | 0.523±0.121† |
| | RandomForest | 0.216±0.089† | 0.298±0.108† | 0.548±0.128† |
| | SVM | 0.228±0.076† | 0.323±0.096† | 0.610±0.129† |
| Deep model | MLP | 0.241±0.077 | **0.351±0.102** | **0.668±0.133** |
| | EEGNet* | 0.123±0.033† | 0.192±0.077† | 0.351±0.088† |
| | RGNN | **0.250±0.087** | 0.347±0.093 | 0.660±0.128 |

ever, even in Stage 2, the classification accuracy remains notable, with a highest accuracy of 25.6% for 80-class classification, 36% for 40-class classification, and an average accuracy of 67.6% for fine-grained 8-class classification. These results indicate that, even when the temporal effect is removed, the collected EEG signals still contain meaningful semantic information, further validating the effectiveness of EEG-ImageNet. The data from Stage 2 will be released in the same format and structure as the original dataset. Since both stages involve the same group of participants, EEG-ImageNet can be used to study the continuity of human encoding of image information across experiments. This multi-stage approach enhances the dataset's potential for exploring more complex cognitive processes, making EEG-ImageNet even more valuable as a research resource.

