# OpenReview forum: "EEG-ImageNet: An Electroencephalogram Dataset and Benchmarks with Image Visual Stimuli of Multi-Granularity Labels"
_ICLR.cc/2025/Conference — Submitted to ICLR 2025_

### Official Review · Reviewer_bg7F · 2024-10-26

**Soundness:** 3
**Presentation:** 2
**Contribution:** 3
**Rating:** 5
**Confidence:** 5

**Summary:**

This article introduces a new EEG dataset recorded through an RSVP (Rapid Serial Visual Presentation) experiment paradigm, where EEG data was collected from 16 participants as they viewed images from the ImageNet dataset. Each participant viewed 4,000 images across 80 categories. The dataset supports multi-granularity analysis, with 40 categories designated for coarse-grained analysis and 40 for fine-grained analysis. The article provides a detailed description of the dataset construction process, including participant selection, stimulus dataset, and acquisition procedure. Additionally, the dataset was used to establish benchmarks for two common image tasks: classification and image reconstruction. Experimental results validate the feasibility of this dataset, providing a foundation for subsequent research.

**Strengths:**

1.This paper makes use of the advantages of EEG and proposes EEG-ImageNet dataset to provide reference for related tasks.

2.The authors provided a detailed description of the experimental acquisition process and preprocessing.

**Weaknesses:**

1.The innovation of this dataset, compared to others, lies in the introduction of multi-granularity labels; however, subsequent experiments did not include a comparison of performance with other models in handling multi-granularity labels.

2.In the benchmarks, the authors chose relatively traditional methods and did not use current SOTA models to validate the dataset.

3.As a new EEG-Image dataset, it has not been compared with previous datasets of the same type, such as THINGS-EEG.

4.In the image classification validation experiments, only Acc was used as the evaluation metric, limiting the effectiveness of the analysis. Additionally, in the image reconstruction task, only four evaluation metrics were employed.

REFERENCES:

[1]Bharadwaj, H. M., Wilbur, R. B. & Siskind, J. M. Still an ineffective method with supertrials/ERPs—comments on “decoding brain representations by multimodal learning of neural activity and visual features”. IEEE Trans. Pattern Anal. Mach. Intell. 45, 14052–14054 (2023).

[2]Grootswagers, T., Zhou, I., Robinson, A. K., Hebart, M. N., and Carlson, T. A. (2022). Human EEG recordings for 1,854 concepts presented in rapid serial visual presentation streams. Scientific Data, 9(1):3.

**Questions:**

1.In Table 2, why does EEGNet perform relatively poorly? On line 378，the authors mentioned that "* indicates the use of time-domain features." What is the significance of this approach?

2.On line 256, it mentions, “The latter 40 categories are designed as a fine-grained task, divided into 5 groups with 8 categories each.” Could this approach lead to a data imbalance issue, making it difficult to directly compare the performance at different granularities in the subsequent image classification task?

---

### Official Review · Reviewer_LG8e · 2024-11-01

**Soundness:** 1
**Presentation:** 4
**Contribution:** 2
**Rating:** 3
**Confidence:** 5

**Summary:**

This paper described a newly collected EEG dataset named EEG-ImageNet, together with two benchmarks which are the object classification and the image reconstruction tasks based on the EEG signal. The dataset is collected by showing images sequentially to the subjects while collecting the EEG signal. The object classification is set to predict the fine-grained and coarse label of the image. And the image reconstruction task is to reconstruct the image based on the EEG signal.

**Strengths:**

1. The overall problem the authors are trying to solve is very important, and building such a dataset will definitely help the area moving forward, as long as the dataset is not wrongly constructed.
2. The benchmarks are set properly, with results verified by several models.
3. The presentation of this paper is generally good.

**Weaknesses:**

I totally understand the efforts the authors put in to collect the dataset and I really appreciate them. However, I have to say that the overall experiment setting to collect the EEG signal with visual stimuli has some intrinsic flaws due to the temporal effect, and can't be made up by simply adding the section A.5.
1. I believe the authors are aware of the temporal effect of RSVP. The general pipeline to show the image sequence is very similar to Spampinato et al.(2017), where the images within the same category are presented in a row to the subject. This is so-called **block design** in a followed paper [1] and is proved to be a wrong way to collect EEG dataset. Shortly speaking, the block design will lead to classifying arbitrary temporal artifacts of the data instead of stimulus-related activity. I verified the claim of [1] in both datasets from Spampinato et al. (2017) and under a controlled setting of our own, and I generally believe the claim is right. Also, the claim is verified by the authors on the proposed dataset in figure 6 where the first few samples in the test set is classified more accurately.
2. While the authors claims that the influence can be mitigated by some techniques including different train-test split or narrower temporal segmentation, it's still a problem. The authors claims a stationary and convergent trend on the last few samples of the test set in their setting, however, it's probably because the temporal influence is convergent to somewhere between this category and the category right after it. To verify this claim, it will be helpful to plot the confusion matrix of the classification benchmark on the dataset. If my guess is correct, we will see a upward trend for the wrongly classified data to be classified to the category right after it. This is also a form of temporal influence that highly biased the model to learn temporal features, and will influence the final result.
3. Let's set apart all the following things and recognize the last sentence of the paper: "the dataset can contribute significantly to future research aimed at mitigating temporal effects in the RSVP paradigm". Then all the proposed two benchmarks are invalid now. Because achieving high results on the two benchmarks encourages a method to actually utilize the temporal effect rather than mitigate the temporal effect, we will have some contradictory directions to optimize on this signal dataset without a clear goal, which will definitely confuse the followed researchers in this direction.
4. By the way, there are some newer datasets proposed but in a more randomized for your information. THINGS-EEG[2] uses a separate stage for validation after all the training data are collected (which is made unaware for the subject), and SEED-DV[3] separate different stages of stimuli and make sure the images of the same stage are only presented in either training or the testing time.

[1] Li R, Johansen J S, Ahmed H, et al. **The perils and pitfalls of block design for EEG classification experiments**[J]. *IEEE Transactions on Pattern Analysis and Machine Intelligence, 2020, 43(1): 316-333.*

[2] Grootswagers T, Zhou I, Robinson A K, et al. Human EEG recordings for 1,854 concepts presented in rapid serial visual presentation streams[J]. Scientific Data, 2022, 9(1): 3.

[3] Liu X, Liu Y, Wang Y, et al. EEG2Video: Towards Decoding Dynamic Visual Perception from EEG Signal[J]. Advances in Neural Information Processing Systems, 2024.

**Questions:**

If you plot the confusion matrix according to the sequence of categories you present to the subjects, what will it look like?

---

### Official Review · Reviewer_YTBj · 2024-11-03

**Soundness:** 3
**Presentation:** 3
**Contribution:** 2
**Rating:** 3
**Confidence:** 5

**Summary:**

EEG-based neural visual representation for image classification tasks is very promising and currently does require more extensive EEG datasets with reliable experiment design to support this research area. The authors provided more experiments/databases in this work than the existing work.

**Strengths:**

EEG-based neural visual representation shows promise for image retrieval and classification tasks, especially in contexts where current datasets are limited in size and have design limitations. This work contributes by providing a more comprehensive EEG dataset tailored for image retrieval and classification, addressing some of these existing challenges.

**Weaknesses:**

The authors missed discussing a significant work, The Perils and Pitfalls of Block Design for EEG Classification Experiments (published in IEEE TPAMI, 2020), which highlights a critical weakness in Spampinato’s 2017 study. Specifically, the block design used in EEG experiments can yield artificially high accuracy levels that are unlikely to translate to real-world testing conditions. Unfortunately, the authors appear to have followed a similar block design in their current study.

While I appreciate that the authors chose a random seed to randomize the order of categories before the experiment, there remains an issue. Once a category is introduced, participants may anticipate that subsequent images within that block are visually similar, which could introduce bias. To improve the experimental design, I think it is essential fully randomizing the presentation of images across all categories rather than within isolated blocks.

**Questions:**

See above. Concern the issues from block design for EEG experiments.

---

### Official Review · Reviewer_MjjY · 2024-11-04

**Soundness:** 3
**Presentation:** 3
**Contribution:** 3
**Rating:** 6
**Confidence:** 4

**Summary:**

In this article, the authors proposed EEG-ImageNet, a dataset with visual stimuli presentation using Rapid Serial Visual Presentation with 16 subjects and a total of 4000 images. The authors present the dataset, classification benchmark and reconstruction studies.

**Strengths:**

The paper is well-written, and tackles a fundamental problem in the EEG-to-IMG field: the lack of datasets.

The authors are precise about the methods used, evaluation within the same subject, and present a 2-step methodology to reconstruct the image given the biosignal.

The overview of the literature on datasets is interesting, although not complete.

**Weaknesses:**

Major:

- The authors fail to list Thing's EEG dataset, version 1 and version 2, for some reason that I don't understand. This dataset is currently widely used in the literature and employs a methodology similar to the one the authors present here. A critical comparison is necessary.

- The results with EEGNet models are very poor, raising a flag about the validation of the dataset for deep learning. I understand that reproducing EEGNet is very challenging. However, more neural networks should be chosen to validate this dataset for deep learning. Some suggestions for classification are: ShallowConvNet, DeepConv, EEGConformer, ATCNet, EEGConformer, EEGNex, and perhaps EEGChannelNet.

- The search for parameters seems to favor traditional machine learning models in this study; a better search for parameters should be made. A good reference to understand the importance of hyperparameters can be found Borra, D. et al (2024).

- Even when training one subject by model, other co-founders are present, as discussed in (Kilgallen et al., 2024; Bharadwaj, 2023; Li et al., 2020; Ahmed et al., 2021). Your paper doesn't discuss these issues, making me question your dataset's applicability. How do you ensure that these confounders aren’t impacting your dataset, too?

Minor:

- The tone of paragraphs 212-215 needs to be reduced; the sentences don't hold up compared to other datasets, Things-EEG, for example.

- Some datasets are missing from table 1, please include them:

Zheng, X., Wang, L., Chen, K., Lyu, Y., Zhou, J., & Wang, L. (2024). EIT-1M: One Million EEG-Image-Text Pairs for Human Visual-textual Recognition and More. arXiv preprint arXiv:2407.01884.

Xu, J., Aristimunha, B., Feucht, M. E., Qian, E., Liu, C., Shahjahan, T., ... & Nestor, A. (2024). Alljoined--A dataset for EEG-to-Image decoding. arXiv preprint arXiv:2404.05553.

Gifford, A. T., Dwivedi, K., Roig, G., & Cichy, R. M. (2022). A large and rich EEG dataset for modeling human visual object recognition. NeuroImage, 264, 119754.

Kaneshiro, B., Perreau Guimaraes, M., Kim, H. S., Norcia, A. M., & Suppes, P. (2015). A representational similarity analysis of the dynamics of object processing using single-trial EEG classification. Plos one, 10(8), e0135697.


References

Borra, D., Paissan, F., & Ravanelli, M. (2024). SpeechBrain-MOABB: An open-source Python library for benchmarking deep neural networks applied to EEG signals. Computers in Biology and Medicine, 182, 109097.

Kilgallen, J. A., Pearlmutter, B. A., & Siskind, J. M. (2024, June). Learning Exemplar Representations in Single-Trial EEG Category Decoding. In 2024 35th Irish Signals and Systems Conference (ISSC) (pp. 1-6). IEEE.

Bharadwaj, H. M., Wilbur, R. B., & Siskind, J. M. (2023). Still an Ineffective Method With Supertrials/ERPs—Comments on “Decoding Brain Representations by Multimodal Learning of Neural Activity and Visual Features”. IEEE Transactions on Pattern Analysis and Machine Intelligence.

Li, R., Johansen, J. S., Ahmed, H., Ilyevsky, T. V., Wilbur, R. B., Bharadwaj, H. M., & Siskind, J. M. (2020). The perils and pitfalls of block design for EEG classification experiments. IEEE Transactions on Pattern Analysis and Machine Intelligence, 43(1), 316-333.

Ahmed, H., Wilbur, R. B., Bharadwaj, H. M., & Siskind, J. M. (2021). Confounds in the data—Comments on “Decoding brain representations by multimodal learning of neural activity and visual features”. IEEE transactions on pattern analysis and machine intelligence, 44(12), 9217-9220.

**Questions:**

The points raised in the weaknesses section

---

### Meta-Review · Area_Chair_JFDe · 2024-12-16

**Metareview:**

This paper proposed a new EEG dataset for image decoding with some baseline methods.

All reviewers raised concerns about the significance of the contribution given the state of the field of EEG image decoding. Experimental results were not considered solid or particularly ambitious making it doubtful about the ability of the field to make real progress based on this work.

**Additional Comments On Reviewer Discussion:**

Reviewers raised concerns about the quality of the results (performance), experimental protocol (parameter selection methodology) and emitted doubts about the quality of the data.

---

### Decision · Program_Chairs · 2025-01-22

Reject